

# You are what you eat: fungal metabolites and host plant affect the susceptibility of diamondback moth to entomopathogenic fungi

Sereyboth Soth[1,2], Travis R. Glare[1], John G. Hampton[1], Stuart D. Card[3], Jenny J. Brookes[1] and Josefina O. Narciso[1]

[1] Bio-Protection Research Centre, Lincoln University, Christchurch, Canterbury, New Zealand
[2] Department of Science, Technology and Innovation Training, National Institute of Science, Technology and Innovation, Chak Angre Leu, Mean Chey, Phnom Penh, Cambodia
[3] Grasslands Research Centre, AgResearch Limited, Palmerston North, Manawatū-Whanganui, New Zealand

Corresponding author
Sereyboth Soth,
sereybothsoth@gmail.com

## ABSTRACT

**Background:** *Beauveria* are entomopathogenic fungi of a broad range of arthropod pests. Many strains of *Beauveria* have been developed and marketed as biopesticides. *Beauveria* species are well-suited as the active ingredient within biopesticides because of their ease of mass production, ability to kill a wide range of pest species, consistency in different conditions, and safety with respect to human health. However, the efficacy of these biopesticides can be variable under field conditions. Two under-researched areas, which may limit the deployment of *Beauveria*-based biopesticides, are the type and amount of insecticidal compounds produced by these fungi and the influence of diet on the susceptibility of specific insect pests to these entomopathogens.

**Methods:** To understand and remedy this weakness, we investigated the effect of insect diet and *Beauveria*-derived toxins on the susceptibility of diamondback moth larvae to *Beauveria* infection. Two New Zealand-derived fungal isolates, *B. pseudobassiana* I12 Damo and *B. bassiana* CTL20, previously identified with high virulence towards diamondback moth larvae, were selected for this study. Larvae of diamondback moth were fed on four different plant diets, based on different types of Brassicaceae, namely broccoli, cabbage, cauliflower, and radish, before their susceptibility to the two isolates of *Beauveria* was assessed. A second experiment assessed secondary metabolites produced from three genetically diverse isolates of *Beauveria* for their virulence towards diamondback moth larvae.

**Results:** Diamondback moth larvae fed on broccoli were more susceptible to infection by *B. pseudobassiana* while larvae fed on radish were more susceptible to infection by *B. bassiana*. Furthermore, the supernatant from an isolate of *B. pseudobassiana* resulted in 55% and 65% mortality for half and full-strength culture filtrates, respectively, while the filtrates from two other *Beauveria* isolates, including a *B. bassiana* isolate, killed less than 50% of larvae. This study demonstrated different levels of susceptibility of the insects raised on different plant diets and the potential use of metabolites produced by *Beauveria* isolates in addition to their conidia.

## INTRODUCTION

The entomopathogenic fungal genus *Beauveria* has been, and continues to be, extensively utilized for the development of biopesticides for insect pest control with more bioinsecticides based on this genus than any other fungal genera (*Faria & Wraight, 2007*; *Mascarin & Jaronski, 2016*). *Beauveria* species are well-suited as the active ingredient within biopesticides due to their ease of mass production, ability to kill a wide range of pest species, consistency in different conditions, and safety with respect to human health (*Zimmermann, 2007*). However, there are still factors limiting the widespread adoption of these fungal-based biopesticides, including slowness to kill target pests, high cost associated with the solid-state production of viable propagules, and variability in performance under some field conditions (*Glare et al., 2012*; *Wraight et al., 2010*). Two under-researched areas, which may limit the deployment of *Beauveria*-based biopesticides, are the type and amount of insecticidal compounds produced by these fungi and the influence of diet on the susceptibility of specific insect pests to these entomopathogens.

*Beauveria* are typically entomopathogenic fungi and their infection cycle, *via* the four main steps of adhesion, germination, penetration, and dissemination is well understood (for a detailed review see *Dannon et al., 2020*). Briefly, conidia of the fungus arrive on susceptible hosts and germinate. An appressorium forms and aided with hydrolytic enzymes, uses mechanical pressure to penetrate all cuticle layers until it reaches the host's haemolymph. A morphogenetic change then occurs whereby filamentous growth ceases and single cells are produced that colonize the host's internal tissues. The fungus can also produce toxic metabolites that help to overcome the insect's immune defense mechanisms leading to insect death (*Dannon et al., 2020*). Further metabolites are produced once the insect dies to inhibit bacterial competition within the insect cadaver before the fungus sporulates on the mummified cadaver of its host, thus disseminating its spores. Some *Beauveria* strains produce several insecticidal metabolites within the host increasing the speed of the infection process ( *Hajek & St. Leger, 1994*).

A great deal of strain variation exists within species of these fungal genera with respect to the amounts and types of toxins produced, including secondary metabolites, cuticle-degrading enzymes, and toxic proteins (*Ortiz-Urquiza et al., 2010*). For example, *in vitro* studies investigating the toxicological properties of *B. bassiana* and *B. pseudobassiana*-derived metabolites revealed that these fungi could produce different amounts of certain metabolite at a range of concentrations depending on the type of media they were previously cultured on (*Berestetskiy et al., 2018*; *Wang et al., 2020*). Therefore, it is likely that genetically distinct isolates of *Beauveria* may kill an insect faster if they are able to produce one or more lethal compound(s) at a relatively high concentration to overcome the host's immune responses.

The current study therefore investigated the toxicity of culture filtrates produced from three genetically diverse strains of *Beauveria* previously identified as highly bioactive towards *Plutella xylostella* (diamondback moth; DBM) larvae (*Soth et al., 2022*). The aim was to identify a *Beauveria* strain that could produce high concentrations of several insect toxins with the rationale that this strain would kill DBM more quickly than less bioactive fungal strains.

Diet may play a significant role in the health and fitness of insect pests. Previous studies showed diets affect susceptibility of insect pests to infection by *Beauveria* fungi (*Santiago-Álvarez et al., 2006*; *Zafar et al., 2016*; *Kangassalo et al., 2015*). Members of the Brassicaceae (commonly referred to as the mustard or crucifer family), which includes important crops such as *Brassica napus* subsp. *napus* (rapeseed), *B. oleracea* (*e.g.*, broccoli, cabbage and cauliflower), *B. rapa* (*e.g.*, turnip) and *Raphanus sativus* (radish), produce glucosinolates within their vegetative organs and seed (*Ishida et al., 2014*). Glucosinolates are a large group of plant-derived secondary metabolites, which contain sulphur and nitrogen, that can impart a bitter flavor when eaten (*Barba et al., 2016*). These metabolites likely function as a plant defense against phytopathogens and invertebrate pests (*Neugart, Hanschen & Schreiner, 2020*) as they are released when the plant is damaged (*e.g.*, by chewing or cutting) (*Bekaert et al., 2012*; *Bidart-Bouzat & Kliebenstein, 2008*; *Hopkins, van Dam & van Loon, 2009*; *Santolamazza-Carbone et al., 2016*). These pungent compounds can act directly as an invertebrate repellent and an antifeedant (*Santolamazza-Carbone et al., 2016*), and indirectly by attracting parasitoids that will feed on invertebrate pests (*Hopkins, van Dam & van Loon, 2009*). Species of Brassicaceae have co-evolved with several invertebrate pests over many millions of years in an arms-race (*Edger et al., 2015*). For example, DBM has developed biochemical adaptations to glucosinolates that allow these insects to feed freely on plants containing these pungent compounds (*Ratzka et al., 2002*). The concentration and type of glucosinolate are largely responsible for determining the plant's susceptibility to invertebrate attack (*Robin et al., 2017*; *Santolamazza-Carbone et al., 2014*). No study has investigated the effect of these compounds on the susceptibility of these pests to entomopathogenic fungi, such as species of *Beauveria*.

## MATERIALS AND METHODS

### Insect diet preparation

Two New Zealand-derived fungal isolates, *B. pseudobassiana* I12 Damo and *B. bassiana* CTL20, previously identified with high virulence towards DBM larvae (*Soth et al., 2022*), were selected for this study. The isolates were sourced from the culture collection held at the Bio-Protection Research Centre (BPRC), Lincoln University, New Zealand.
The isolates were maintained as axenic cultures on PDA (Thermo Fisher Scientific, Waltham, MA, USA; Oxoid Ltd., Basingstoke, UK) at 22 °C until required. The harvest and preparation methods for *Beauveria conidia* were previously described in *Soth et al. (2022)*.

Broccoli, cv. de Cicco, cauliflower, cv. All Seasons; and radish, cv. Red Cherry (McGregor's, AHM Ltd., Auckland, New Zealand); and cabbage, cv. Arisos NS (South Pacific Seeds Ltd., Pukekohe, New Zealand) were used in this study. For each of the four

cultivars, twenty four 1L pots were filled with a mixture of 400 kg of potting mix + 100 kg of pumice + 500 g of agricultural lime + 500 g of Hydraflo + 1,500 g of Osmocote Exact, Standard 3-4M (16-3.9-10+1.2 Mg+TE) (Scott's Osmocote®). Ten seeds were then sown per pot. Plants were grown in a glasshouse (average temperature during the experiment was 23 ± 5 °C) and watered twice a day for 6 weeks before the plants were transferred to a controlled temperature room set at 25 ± 2 °C. Four plants of each cultivar, from four different pots, were randomly selected for glucosinolate analysis. Glucosinolates were extracted using the method described by *Doheny-Adams et al. (2017)* while analysis was undertaken according to the procedure of *Mawlong et al. (2017)*. Briefly, extracted solutions from four young leaves (the third leaf counted from the bud) and four old leaves (the sixth leaf counted from the bud) per plant were used for glucosinolate content comparison while a spectrophotometer (GENESYS™ 10 Series, Thermo Fisher Scientific Corp., Singapore) was used for glucosinolate analysis. The linear regression involving sinigrin standard solutions was described by *Hu et al. (2010)*.

Five pots containing 10 plants each were transferred to a cage made of fibre net (mesh size 1 mm × 1mm). A total of 12 cages were used for these two assays (four cages for colony establishment, four cages for the first assay, and four cages for the second assay). Around 100 newly emerged, active DBM moths were collected and the subsequent DBM larvae were fed on 6-week-old cabbage cv. Arisos NS and kept within a controlled temperature room (25 ± 2 °C) with a 12:12 day: night light period. The next generation of DBM larvae were maintained on the plant species selected for the experiments until they developed into adults. These adult moths were transferred to the same plant cultivar to start new colonies. Third instar larvae hatching from the second (first assay) and third generations (second assay) were collected from plants belonging to the four different plant species.

## Supernatant preparation

Three genetically diverse isolates of *Beauveria* (*B. pseudobassiana* FRhp, *B. pseudobassiana* FW Mana, and *B. bassiana* CTA20) were selected from the BPRC culture collection according to their previous pathogenicity to DBM larvae (*Soth, 2021*). *B. pseudobassiana* FW Mana had previously shown a faster DBM kill rate than other related strains but was slow to sporulate on DBM cadavers, indicating potential toxin involvement due to incomplete colonization of the cadaver. Isolates *B. pseudobassiana* FRhp (which showed almost the same speed of kill and percentage sporulation on cadavers as FW Mana) and *B. bassiana* CTA20 (which killed slower but had a higher rate of sporulation on cadavers) were used for comparison. The three *Beauveria* isolates were cultured on PDA (Thermo Fisher Scientific, Waltham, MA, USA; Oxoid Ltd., Basingstoke, UK) at 23 ± 1 °C 12:12 (D: L) until use.

Conidial suspensions of the selected *Beauveria* isolates were prepared as described earlier. One hundred microliters of each conidial suspension were inoculated into 100 ml of PDB within a 250 ml Erlenmeyer flask. Flasks were incubated at 24 ± 1 °C on an orbital shaker (Amerex Instruments, Inc, Concord, CA, USA) set at 150 rpm under a photoperiod of 12:12 D/L for 7 days. The subsequent suspension was filtered *via* a pre-sterilized vacuum-driven disposable filtration system (0.22 μm, Millipore, Express™

PLUS, Stericup and Steritop, Darmstadt, Germany) to remove fungal mycelium. The resulting culture filtrates were kept at 4 °C until use. Full-strength culture filtrates were diluted with 0.01% Triton™ X-100 50:50 to obtain half-strength solutions. Third instar DBM larvae were used in all bioassays. These larvae were obtained from a laboratory colony maintained at 25 ± 2 °C, 12:12 D/L and fed on cabbage leaves cv. Arisos NS at the BPRC.

## Bioassay protocols

Bioassay procedures including Petri dish preparation, spraying methods, incubation conditions, larval mortality assessment, and mycosed cadaver assessment were previously described in *Soth et al. (2022)*. Additionally, for the supernatant assay, the spray program was from half to full strength, and the incubation trays did not require covering with plastic bags to prevent moisture loss. Assay experiments were carried out twice. Petri dishes on trays within the incubator were randomized every day (Fig. S1).

## Statistical analysis

Genstat 20th Edition was used to analyze the glucosinolate data using ANOVA (*Genstat Committee, 2021*). For estimation of $LC_{50}$, the total mortalities were calculated using Probit analysis (*Finney, 1971*), using the log10 of the doses, and the logistic model. Control mortality was estimated, with separate $LC_{50}$ and slope parameters for each species and isolate. This was done by fitting separately to each species by isolate combination with the control included: the command as implemented requires a separate control for each species by isolate combination to allow all treatments to be analyzed simultaneously. Less than 20% mortality in the control treatment was acceptable (*Glare et al., 2008*), or otherwise, the assay was repeated. Abbott's formula was used to correct for the control mortality (*Abbott, 1925*). For both trials, the total mortality for each replicate of each treatment was calculated. The final percentage mortalities were analyzed using a binomial generalized linear model with a logit link (*McCullagh & Nelder, 1989*). The median lethal time ($LT_{50}$) was estimated for each treatment using the Kaplan-Meier approach (*Kalbfleisch & Prentice, 2011*).

# RESULTS

## Insect diet assay

The total glucosinolate content from young leaves of the four plant species did not differ significantly ($F_{3,3} = 2.22$, $p = 0.156$). However, from old leaves, radish contained a higher ($F_{3,3} = 15.30$, $p < .001$) glucosinolate content than the other three species. Only cabbage and radish differed significantly ($F_{3,3} = 6.85$, $p = 0.011$) with respect to average (old and young leaves) glucosinolate content (Table 1).

There were no significant differences ($F_{2,3} = 1.77$, $p = 0.191$) among the mortality rates of larvae fed on the four species of Brassicaceae for either of the *Beauveria* strains at the medium or high application rates (Table 2). There were no significant differences among the mortality of DBM across the four Brassicaceae species at the low application rate for I12 Damo. However, there were significant differences observed at the low application rate

**Table 1 Mean glucosinolate content (±SE) of young and old leaves of four Brassicaceae species.**

| Mean ± SE (mg/mL) | | | |
|---|---|---|---|
| Species | Young leaves | Old leaves | Average |
| Broccoli | 0.24 ± 0.068 | 0.26 ± 0.010a | 0.25 ± 0.039ab |
| Cabbage | 0.19 ± 0.043 | 0.21 ± 0.007a | 0.20 ± 0.025a |
| Cauliflower | 0.25 ± 0.059 | 0.24 ± 0.023a | 0.25 ± 0.041ab |
| Radish | 0.30 ± 0.057 | 0.32 ± 0.025b | 0.31 ± 0.041b |
| $F_{3, 3}$ | 2.22 | 15.30 | 6.85 |
| $p$-value | 0.156 | <0.001 | 0.011 |

Note:
Values followed by the same letter are not significantly different ($p > 0.05$) according to Tukey's honestly significant difference (HSD) test.

**Table 2 Mortality (% ± SE) of diamondback moth (DBM) larvae fed on different species of Brassicaceae after correcting for control mortality using Abbott's formula caused by spraying conidia of both *Beauveria pseudobassiana* isolate I12 Damo and *Beauveria bassiana* isolate CTL20.**

| Species and isolate code | Mean larval mortality at different conidial application rates (conidia/spray) | | | |
|---|---|---|---|---|
| | Species | Low ($6 \times 10^4$) | Medium ($6 \times 10^6$) | High ($6 \times 10^7$) |
| *B. pseudobassiana* I12 Damo | Broccoli | 78 ± 7.00 | 94 ± 5.55 | 100 ± 0.00 |
| *B. pseudobassiana* I12 Damo | Cabbage | 40 ± 5.00 | 100 ± 0.00 | 100 ± 0.00 |
| *B. pseudobassiana* I12 Damo | Cauliflower | 61 ± 6.00 | 94 ± 5.55 | 100 ± 0.00 |
| *B. pseudobassiana* I12 Damo | Radish | 56 ± 5.00 | 94 ± 5.55 | 100 ± 0.00 |
| $F_{2, 3}$ | | 6.01 | 1.00 | NA |
| $p$-value | | 0.087 | 0.500 | NA |
| *B. bassiana* CTL20 | Broccoli | 33 ± 11.00 | 83 ± 4.00 | 94 ± 6.00 |
| *B. bassiana* CTL20 | Cabbage | 45 ± 5.55 | 94 ± 6.00 | 94 ± 6.00 |
| *B. bassiana* CTL20 | Cauliflower | 39 ± 4.00 | 100 ± 0.00 | 100 ± 0.00 |
| *B. bassiana* CTL20 | Radish | 61 ± 3.00 | 100 ± 0.00 | 100 ± 0.00 |
| $F_{2, 3}$ | | 9.22 | 8.03 | 1.00 |
| $p$-value | | 0.050 | 0.060 | 0.500 |

for isolate CLT20 which killed more DBM larvae fed on radish compared to those fed on broccoli (Table 2).

There were no significant ($F_{2,3} = 1.21$, $p = 0.428$) differences among the estimated application rates of isolate I12 Damo to reach an $LC_{50}$ of DMB larvae fed on the different Brassicaceae species (Fig. 1). There were, however, significant differences with respect to isolate CLT20 that required a significantly higher rate of application to reach the $LC_{50}$ with DBM fed on cabbage and broccoli compared to larvae fed on radish (Fig. 1).

Larvae raised on broccoli, cauliflower, and radish reached the $LT_{50}$ in less than 7 days at the low application rate, while for those on cabbage, took longer. At the medium application rate, larvae fed on cabbage and cauliflower took less than 3 days to achieve $LT_{50}$, and for those on broccoli and radish, took less than 4 days to reach 50% mortality. Larvae fed on all four species of Brassicaceae had 50% mortality after around 2 days at the

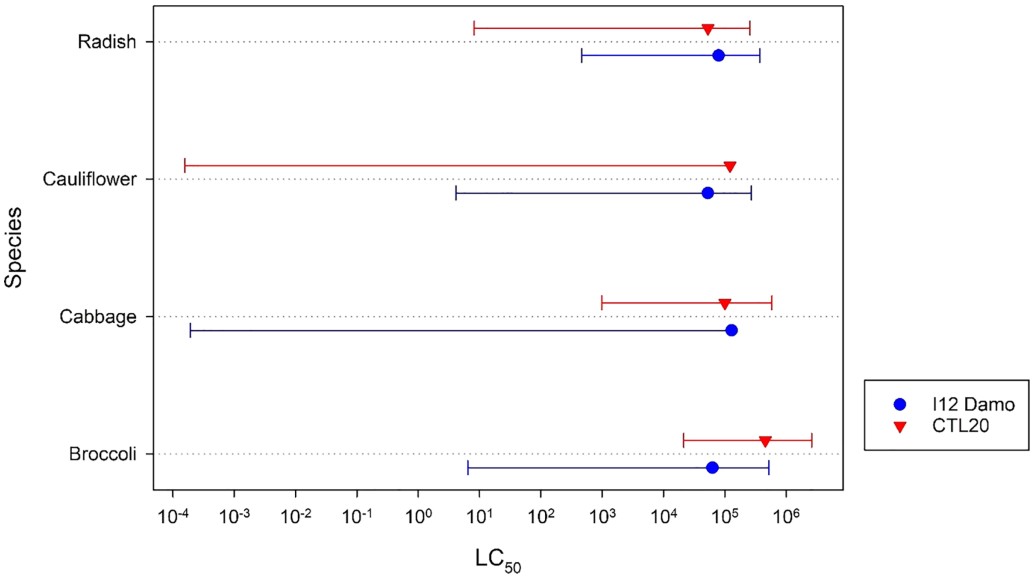

**Figure 1 Estimated application rate of *Beauveria pseudobassiana* isolate I12 Damo and *Beauveria bassiana* isolate CTL20 required to kill 50% of DBM larvae fed on four different species of Brassicaceae.** Error bars are 95% confidence limits.

high application rate (Fig. 2). There was no statistical difference among the four species of Brassicaceae ($F_{2,3}$ = 2.21, $p$ = 0.120).

For isolate CTL20, only larvae raised on cabbage and radish achieved $LT_{50}$ within 7 days at the low application rate. Within the group, there were significant differences between radish and cauliflower ($F_{2,3}$ = 24.54, $p$ < 0.001). $LT_{50}$ took less than 5 days at the medium application rate and less than 4 days at the high application rate across all four brassicas (Fig. 2). There was no significant difference among the four species of Brassicaceae at the medium ($F_{2,3}$ = 1.49, $p$ = 0.353) and the high application rates ($F_{2,3}$ = 0.45, $p$ = 0.772).

## Percentage of cadavers that supported sporulation

The percentage of cadavers that supported sporulation of isolate I12 Damo was statistically significantly different at the low application rate among larval diets ($F_{2,3}$ = 19.08, $p$ = 0.007). Cadavers of larvae fed on broccoli and radish supported less sporulation than those raised on cabbage and cauliflower, while those raised on cabbage supported more sporulation than those raised on cauliflower. There were no significant differences within the four species of Brassicaceae at the medium and high rates (Fig. 3A). Percentage sporulation results for isolate CTL20 did not differ significantly among all rates and the four varieties of Brassicaceae (Fig. 3B).

## Supernatant assay

Application of the filtered supernatants from the three *Beauveria* isolates, at the two different concentrations (half and full-strength), resulted in 20% to 65% mortality of DBM larvae within 7 days post-treatment. The application of *B. pseudobassiana* FW Mana supernatant resulted in higher mortality of DBM than the other two isolates at both concentrations. At the half-strength, *B. pseudobassiana* FW Mana killed 35% and 20%

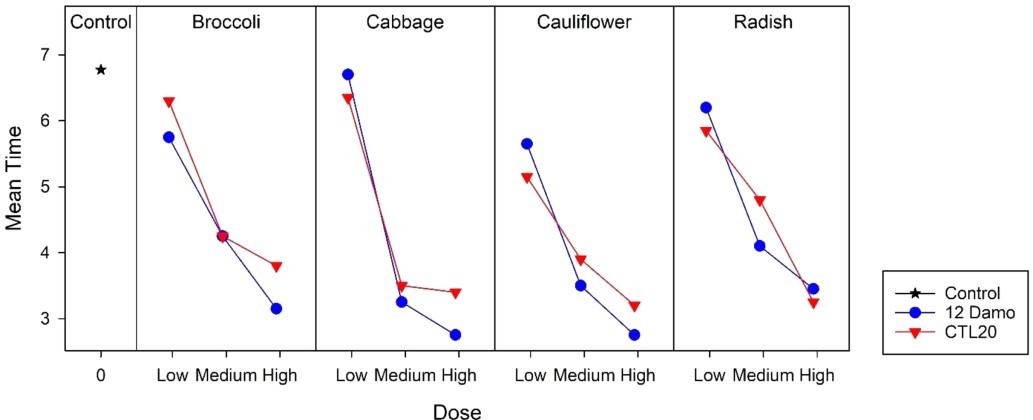

**Figure 2** $LT_{50}$ of DBM larvae fed on four species of Brassicaceae across three application rates of *Beauveria pseudobassiana* isolate I12 Damo and *Beauveria bassiana* isolate CTL20 estimated using the Kaplan-Meier method.

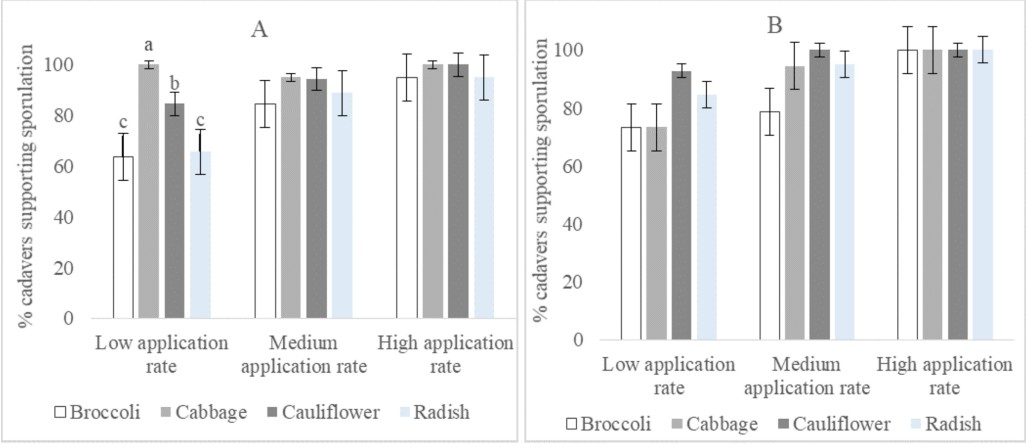

**Figure 3** Percentage of DBM cadavers that supported sporulation after larvae were fed on four brassicas treated with three rates of *Beauveria pseudobassiana* isolate I12 Damo (A) and *Beauveria bassiana* isolate CTL20 (B).

more DBM larvae than isolates *B. pseudobassiana* FRhp and *B. bassiana* CTA20, respectively, which was statistically significant ($F_{2,2} = 71.75$, $p < 0.003$). At the full-strength, *B. pseudobassiana* FW Mana killed 25% more DMB larvae than both FRhp and CTA20, which was also statistically significant ($F_{2,2} = 113.56$, $p < 0.001$) (Fig. 4).

The median time to die could not be estimated in all but two cases (both doses for FW Mana, with medians of 7 and 6 days), and the upper confidence limit could not be estimated for any case using Kaplan-Meier estimates. This is because the final percentage of dead larvae was below 50% for all except FW Mana.

## DISCUSSION

In this study, we examined two areas of research that will advance the deployment of *Beauveria*-based biopesticides, namely the effect of insect diet and the use of culture filtrates produced by these fungi. We utilized two previously identified highly virulent

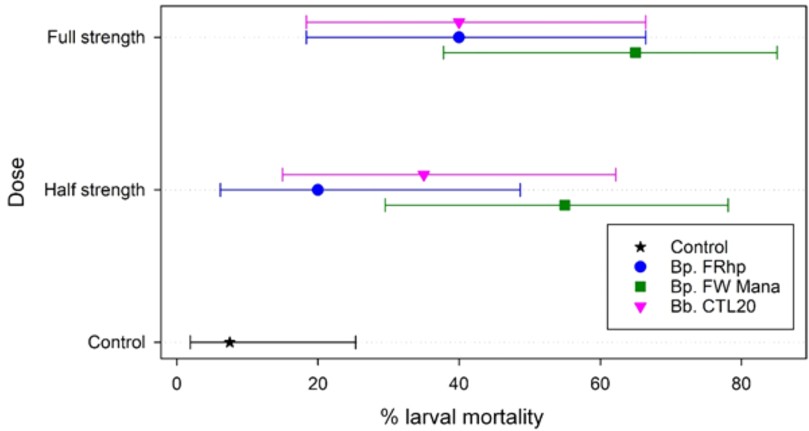

**Figure 4 The percentage mortality of DBM larvae using fungal supernatant of two *Beauveria* isolates applied at two different concentrations.** Error bars are 95% confidence limits.

isolates of *Beauveria* (*Soth, 2021*) and studied the effect of diet on the mortality of DBM, an important agricultural pest. In this study, radish had a higher glucosinolate content than broccoli, cabbage, and cauliflower, but only from old leaves. Insect-resistant properties of Brassicaceae species are related to the glucosinolate metabolites rather than the total amount of glucosinolates in the plant (*Robin et al., 2017*; *Santolamazza-Carbone et al., 2014*; *Sarosh et al., 2010*). At least 120 of these metabolites have been reported in many members of the Brassicaceae family (*Fahey, Zalcmann & Talalay, 2001*). Among these 120 metabolites, three hydrolysis compounds, isothiocyanates, thiocyanates, and nitriles, are known to play a significant role in plant protection against herbivores (*Halkier & Gershenzon, 2006*).

For the four Brassicaceae species assessed within the current study, radish had a higher glucosinolate content than cabbage. However, previous Korean studies on glucosinolate content of eight Brassicaceae species, found that cauliflower and radish contained lower quantities of glucosinolates than the others (*Hwang et al., 2019*; *Kim, Seo & Ha, 2020*). This difference in results may be explained by cultivar differences or environmental effects. For example, several studies have shown that cultivars of cabbage can exhibit significantly different amounts of glucosinolates (*Hamilton et al., 2005*; *Robin et al., 2017*, *2016*). The New Zealand cultivars used in the present study differed from those used in South Korea (*Hwang et al., 2019*; *Kim, Seo & Ha, 2020*). Additionally, brassica crops produced more glucosinolates in response to abiotic stresses and the damage by herbivorous pests, including DBM (*Antonious, Bomford & Vincelli, 2009*). This study showed the age of plant could also influence glucosinolate content.

The susceptibility differed for *B. bassiana* CTL20, which produced higher mortality of DBM larvae raised on radish at the low application rate, while the two higher rates showed no increase in DBM mortality among the four Brassicaceae species used in this study. DBM larvae fed on this high glucosinolate brassica were more susceptible than those fed on a low glucosinolate brassica to the isolate *B. bassiana* CTL20. A previous study on food utilization and consumption of DBM using eight cultivars of cabbage showed no
relationship between low and high glucosinolate groups (*Karmakar, Pal & Chakraborty, 2021*), indicating that DBM can feed on high and low glucosinolate plants without adverse effects on health fitness. However, another study showed that cabbage cultivars containing low glucosinolates were more susceptible to DBM larvae than those containing higher glucosinolates (*Robin et al., 2017*). The result for *B. bassiana* CTL20 in this study showed that infection by this fungal isolate might be related to glucosinolate content, as DBM larvae fed on a higher glucosinolate containing-brassica were more susceptible to fungal infection than those raised on lower glucosinolate plants. This result aligned with previous results that found high glucosinolate containing-plants caused larvae to be more vulnerable to the infection by insect pathogens (*Hopkins, van Dam & van Loon, 2009*; *Robin et al., 2017*; *Santolamazza-Carbone et al., 2016*). As previously noted the total amount of glucosinolates did not determine plant responses to DBM damage, but rather it was some specific glucosinolates (*Robin et al., 2017*). In this regard, although radish had a higher quantity of glucosinolates, differences may be due to specific compounds rather than the total. Further work is required to determine which glucosinolates are present in this radish cultivar. Therefore, further work would aim to identify the types of secondary metabolites produced by these species of Brassicaceae.

The percentage of DBM cadavers that supported sporulation from *B. pseudobassiana* I12 Damo applied at a low application rate to the DBM larvae for the four diets differed significantly. No difference was found for *B. bassiana* CTL20 among larvae fed on the four Brassicaceae species. Glucosinolate metabolites may inhibit the growth of fungi (*Teng et al., 2021*). These metabolites may have reduced the percentage sporulation of isolate *B. pseudobassiana* I12 Damo on infected cadavers. Three-way interactions among plants, arthropods, and entomopathogenic fungi contribute significantly to the development of each dimension (*Biere & Tack, 2013*). For example, the production of newly formed conidia of *B. bassiana* was considerably higher on cadavers of sweet potato whitefly fed on melon than on cotton (*Santiago-Álvarez et al., 2006*). A mixture of leaf extract using two brassicas in a cultivating medium improved the radial growth of *Beauveria* isolates compared to the control (*Cerritos-Garcia et al., 2021*). At the low application rate, the percentage of cadavers supporting sporulation of isolate *B. pseudobassiana* I12 Damo was greater on DBM fed on cabbage and cauliflower than on broccoli and radish. It is possible that glucosinolate causes modifications to the microbiome of the insect stomach, and these modifications increase the larvae's resistance to fungus infection. Because too many conidia overwhelm the insect defenses too fast, this priming impact on insect defenses only becomes apparent when low concentrations of conidia are administered (*Sontowski et al., 2022*). An efficacy assessment of *B. bassiana* in controlling two-spotted spider mite on five species of plants found mortality was high on beans and cucumber, but conidia viability was the same, as was persistence (*Gatarayiha, Laing & Miller, 2010*). This finding was similar to isolate *B. bassiana* CTL20 that showed a significant difference in mortality among the larvae from the four brassicas tested. Consideration could be given to using isolates, such as *B. pseudobassiana* I12 Damo, to control DBM larvae damage on broccoli and cabbage, and isolates such as *B. bassiana* CTL20 to control DBM larvae damage on cauliflower and radish because they require a lower application rate and shorter time to
death than reported for previous studies (*Batta et al., 2010*; *Furlong, 2004*; *Kirubakaran et al., 2014*; *Medo et al., 2021*; *Nithya et al., 2019*). Another technique to improve DBM control could be integrated fungal application with natural insecticides based on glucosinolate metabolites. Some glucosinolate profiles, particularly isothiocyanates, improved toxicity to brassica specialists, *P. xylostella* (DBM) and *Pieris rapae* (White cabbage butterfly) through myrosinase activities (*Agrawal & Kurashige, 2003*; *Li et al., 2000*). Thus, a biopesticide based on isothiocyanates may be potentially incorporated with a fungal spray to optimize the application.

The extracted actives assay showed that the culture filtrates of *B. pseudobassiana* FW Mana achieved higher mortality for DMB larvae than previously published studies using extractions. For example, culture filtrates of *B. bassiana* Bb-2 from China achieved accumulative mortality for DBM larvae of only 37% (*Gao, Hu & Wang, 2012*), which was similar to that from the use of supernatant from *B. bassiana* CTA20 in the current study. This current result is the first report to show that filtered supernatant of an isolate of *B. pseudobassiana* killed more DBM larvae than did a *B. bassiana* isolate. A study investigating the insecticidal activity of *B. pseudobassiana* and *B. bassiana* towards *Spodoptera littoralis* (cotton leafworm) identified fungal isolates that achieved mortalities of up to 67%, while one of the *B. bassiana* isolates did not kill any insects (*Resquín-Romero, Garrido-Jurado & Quesada-Moraga, 2016*). Herein, at least one isolate of *B. pseudobassiana* produced a higher amount of insecticidal toxins, secondary metabolites and/or with activity enzymes against DBM larvae than *B. bassiana*.

Investigation of the insecticidal activity of *B. bassiana* towards green peach aphid (*Myzus persicae*) showed that the chosen isolates achieved 15% to 79% aphid mortality (*Kim et al., 2013*). A recent study using supernatants of *B. bassiana* against *M. persicae* found mortality of between 30% and 70% when extracting from different media and at different fermentation times (*Cheong et al., 2020*). In another study, aphid mortality was higher than DBM mortality, which was up to 48% using the filtrated supernatant of *B. bassiana* Bb-2 (*Gao, Hu & Wang, 2012*). *Beauveria* metabolites may only have a mode of action through the digestive pathway, and DBM larvae can develop resistance to many insecticidal compounds. In New Zealand, a biopesticide producing company showed that a combination of insect toxins and conidia within a spray led to optimization of efficacy (*Glare & O'Callaghan, 2019*). Thus, the application of *B. pseudobassiana* FW Mana may also be improved in this manner. *B. pseudobassiana* could kill DBM larvae through insect toxins more effectively than the *B. bassiana* isolates tested. Previously, an assessment of toxin profiles between these two species revealed differences (*Wang et al., 2020*), some of which may assist in overcoming the immune system of a particular insect. Two enzymes, chitinase and protease, have been found to have dual functions: aiding fungal infection and toxicity to some insects (*Kim et al., 2013*; *Montesinos-Matías et al., 2011*). However, a study showed no correlation between their production and mortality, and the enzymes did not affect the fungal ability to kill the green peach aphid with metabolites from an isolate of *B. bassiana* (*Cheong et al., 2020*). The aphicidal compounds have yet to be identified.

The median lethal time ($LT_{50}$) of the supernatant from the three isolates ranged from 5.4 to 13.12 days, with only FW Mana achieving an $LT_{50}$ within 7 days (5.4 days for full-

strength, 6.42 days for half-strength). Previously, DBM mortality was 27% and 36% for 1 and 2 days after spraying, respectively, when testing with toxins of *B. bassiana* Bb-2 (*Gao, Hu & Wang, 2012*). At 25 °C, $LT_{50}$ was longer than 7 days when testing extracted toxins of two *B. bassiana* isolates against *Spodoptera litura* (*Herlinda et al., 2020*). When using extracted supernatant of *B. bassiana* isolates on green peach aphid, mortality reached 78.6% within 2.7 days after treatment for the most effective isolate, while the less effective isolates required 9.8 days (*Kim et al., 2013*). When using crude protein extracted from *I. fumosorosea* to assay on DBM, the $LT_{50}$ was 6 days post-treatment but earlier with a protein concentration, which may enhance the antifeedant characteristic (*Freed et al., 2012*). Spraying *S. littoralis* using extracted supernatant of a *B. pseudobassiana* isolate achieved $LT_{50}$ by day 4 after treatment, while a *B. bassiana* isolate had no effect (*Resquín-Romero, Garrido-Jurado & Quesada-Moraga, 2016*). However, the study found promising results when combining both conidia and extracted toxins in the spray, giving 100% mortality within four to 5 days after application.

## CONCLUSIONS

Conidia are generally the most infective propagules from entomopathogenic fungi and most *Beauveria*-based biopesticides are based on the asexual, non-motile conidia. *Beauveria* conidia can germinate directly on a target insect, with the resulting germ tube able to directly penetrate the insect's protective cuticle before the fungus colonizes and kills the host. This process is therefore not reliant on the target pest ingesting the active ingredient of the pesticide (*e.g.*, propagules of the fungal entomopathogen) as is the case with many systemic chemical insecticides (*Chandler, 2017*). Isolate *B. pseudobassiana* I12 Damo and *B. bassiana* CTL20 showed different effects on DBM larvae when fed on different diets. When feeding on broccoli and cabbage, DBM larvae were more susceptible to infection by isolate *B. pseudobassiana* I12 Damo. Conversely, those larvae raised on cauliflower and radish were more vulnerable to infection by isolate *B. bassiana* CTL20. The use of these two isolates should consider their performance based on glucosinolate contents of each Brassicaceae variety (if lower, apply *B. pseudobassiana* I12 Damo, if higher, apply *B. bassiana* CTL20). This study showed the potential of *Beauveria* toxins to assist the control of DBM. The isolate *B. pseudobassiana* FW Mana is the most suitable candidate for managing this insect using extracted toxins. Testing of FW Mana isolate toxin on other insect pests and under diverse conditions will be necessary to confirm the toxin efficacy and consistency, as well as non-host safety. Combining conidia and extracted toxins of the same isolates may potentially give an additional effect or synergistic response for DBM control.

## ACKNOWLEDGEMENTS

We thank David Saville (Saville Statistical Consulting Limited), David Baird (Statistical Consultant VSN (NZ) Limited and Genstat Developer), and Ruth Butler (Statistical Consultant, StatsWork 2022 Ltd) for statistical advice.

### Funding

Sereyboth Soth was supported by a New Zealand Ministry of Foreign Affairs and Trade scholarship. Additional funding was supplied by Lincoln University. The funders had no role in study design, data collection and analysis, decision to publish, or preparation of the manuscript.

### Grant Disclosures

The following grant information was disclosed by the authors:
New Zealand Ministry of Foreign Affairs and Trade Scholarship.
Lincoln University.

### Competing Interests

Stuart D. Card is employed by AgResearch Limited. The authors declare that they have no competing interests.

### Author Contributions

- Sereyboth Soth conceived and designed the experiments, performed the experiments, analyzed the data, prepared figures and/or tables, authored or reviewed drafts of the article, and approved the final draft.
- Travis R. Glare conceived and designed the experiments, performed the experiments, analyzed the data, prepared figures and/or tables, authored or reviewed drafts of the article, and approved the final draft.
- John G. Hampton conceived and designed the experiments, performed the experiments, prepared figures and/or tables, authored or reviewed drafts of the article, and approved the final draft.
- Stuart D. Card conceived and designed the experiments, performed the experiments, prepared figures and/or tables, authored or reviewed drafts of the article, and approved the final draft.
- Jenny J. Brookes conceived and designed the experiments, performed the experiments, prepared figures and/or tables, and approved the final draft.
- Josefina O. Narciso conceived and designed the experiments, performed the experiments, prepared figures and/or tables, and approved the final draft.

### Data Availability

The raw data is available in the Supplemental Files.

### Supplemental Information

Supplemental information for this article can be found online at http://dx.doi.org/10.7717/peerj.14491#supplemental-information.

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
