# Peer review of "You are what you eat: fungal metabolites and host plant affect the susceptibility of diamondback moth to entomopathogenic fungi"

_PeerJ, doi:10.7717/peerj.14491_

## Round 0.1 · original submission · Major Revisions

Dear authors, all reviewers found your work interesting and with informative results. However, they pointed out some weaknesses that need to be addressed. Please, pay attention specially to comments raised by reviewer 1, and revise all statistics analyses performed and provide a better description of the methods and experimental set up.

Reviewer 1 ·

Basic reporting

The manuscript is interesting and relevant. The language is clear, and the structure is according to the journal.
The title should be modified because tends to create confusion and it is sensational. I sugesst to the authors remove the first part and replace “insect diet” by “host plant” (Fungal metabolites and host plant affect the susceptibility of diamondback moth to entomopathogenic fungi).
The introduction shows properly the problem, but not only Beauveria strains produce metabolites within the host (lines 73-74 and lines 76-80). Please rewrite all this part including other genera that also produce metabolites as virulence factors.
Figures should be changed. Figures 1 and 2 may be a new figure 1 composed by A) and B), because both are similar with different strains. Figures 3 and 4 may be a new figure 2. And figures 5 and 6 may be removed from the manuscript according to a comment in experimental design section.

Experimental design

In general, it is well structured and coherent, but there is two main points that should be adressed.
1.- Metabolite assay subheading is really two different experiments. The first one is performed with fungal extracts and the second one is using suspensions. The first experiment is well conducted, but the second one should be removed. The killed larvae were not surface disinfected and therefore the authors cannot be sure that the fungus comes from inside the insect. Please also remove lines 258-270 from results and figures 5 and 6.
2.- The authors support the LT50 formula with a personal communication or Soth et al (2022) which is also supported by a personal communication. Please give more information about the origin of the formula with a new reference. On the other hand, the standard errors of LT50 are not supported by checking the excel files give by the authors. Please clarify if LT50 is calculated for the whole sample or for every repetition. If the authors finding it difficult to suppor the formula, there is more common ways of calculating LT50 like Throne et al. 1995 (Throne, J. E., D. K. Weaver, V. Chew, and J. E. Baker. 1995. Probit analysis of correlated data: Multiple observations over time at one concentration. J. Econ. Entomol. 88: 1510-1512). Or even alternatives relatively similar like Median lethal time (MST) from Kaplan-Meier survival analysis.
Other minor points to address are:
Lines 124-125. The sentence should be rewritten because the spores cannot be virulent up to six months in a suspension. As soon as the suspension is performed the conidia start the process of germination.
Line 173. Were the culture filtrates standarized? There were for sure differences between them, so it is difficult to make comparisons between them.
Lines 208-209. Please indicate that ANOVA analysis is only for glucosinolate content. Besides, ANOVA should not be used to analyzed mortality data because they are binary data. The correct analysis for them is a Generalized Linear Model with a binomial distribution. After reading the manuscript I am confused regarding the type of analysis used in each experiment. Please clarify this issue.

Validity of the findings

Degree of freedom in Result heading are not supported by Material and methods and excel files. For example, in line 230 if the authors are analysing the data of 4 species of plants the degree if freedom is 3 (n-1) instead of 2.
Line 315. Please remove the dot after DBM.
Discusion and conclusions about sporulation cadavers sould be removed.

Reviewer 2 ·

Basic reporting

I appreciated the opportunity to review this manuscript titled “You are what you eat: fungal metabolites and insect diet affect the susceptibility of diamondback moth to entomopathogenic fungi”. The current manuscript shows an innovative, interesting, and reliable method to increase the efficiency of the entomopathogenic fungi Beauveria spp. against the diamondback moth in combination with other compounds and the influence of pest diet on the success of the fungi. However, there are some points through the paper that must be improved before publication. For that, I will recommend a major revision of the manuscript. The paper is well written and English language used is clear and professional. Introduction and discussion are good in structure, length, and content.

Experimental design

I only have some general comments for the authors to consider for potential to strengthen this report, particularly in methods and results, which must be reviewed and improved to clarify some points of the procedures performed:
- Authors are considering only secondary metabolites from Beauveria spp. but what about other compounds? Some strains Beauveria strains (as well as other entomopathogenic fungi) can secret other insecticidal compounds such as proteins (Ortiz-Urquiza et al., 2010, Journal of Invertebrate Pathology, 105: 270-278).
- Why the authors used different strains for pathogenicity and metabolite bioassays? Authors should clarify with they didn’t use same strains for all experiments. It seems more logical use the same ones in the current context.
- Bioassays, in general, must be rewritten to explain better the procedures performed. Right now, it is a bit confusing and some relevant information (repetitions, number of DBM per repetition, etc.) are missing. Maybe authors could support the explanation of the experiments with a figure-scheme.
- How authors obtained secondary metabolites from Beauveria is not very clear in methods. Please, authors should considerer to better explain the steps and also separate it in a new section.

Validity of the findings

I found analyses in general are weak. Authors have not provided Probit outputs in an appropriate way. It is important to include the full analysis (regression, X2, slopes, etc) to confirm that calculations were correct. However, three different dosages are very few to get a good adjustment of Probit regression. I recommend the authors to calculate mean survival times using Kaplan-Meyer or binomial analysis for mortalities (depends on the number of repetitions performed and not mentioned). F and p values from ANOVAs are provided in results and also in tables, which is redundant. It will be more appropriate do not include F and p values in tables. Then, authors could add the significance of the analysis (letters) even if there were not significant results.

Reviewer 3 ·

Basic reporting

In this manuscript, Soth et al. describe how the susceptibility of the DBM to fungal infection may depend on diet. In addition, they report that the cell-free culture supernatants of three strains of Beauveria sp. cause similar mortalities in DMB as conidia of Beauveria applied at a low dose (4 X 10^6 conidia/mL). Overall, the paper reads very well, and the experimental design is solid. Also, the results are well described and integrated with the existing literature.

Experimental design

The research is within the aims and scope of the journal. The research question is well defined. Solid experimental design and data analysis. Methods well described.

Validity of the findings

By looking at the data, it is clear that the insect diet affects the susceptibility to fungal infection. However, this effect is only noticeable when the fungus is applied at a low dose. Although such an effect could be linked somehow to the glucosinolate content or the type of glucosinolate in the diet, the authors do not show a direct connection or effect between glucosinolate and fungal infection. Thus, saying that the lower mortality and sporulation observed with certain tested strains of Beauveria could be due to an antifungal effect of glucosinolates is speculative. Is there any proof that ingested glucosinolates accumulate in the haemolymph and can impair fungal growth within the haemocoel? Perhaps, glucosinolate induces changes in the microbiome of the insect gut, and these changes make the larvae more resistant to fungal infection. Thus, this priming effect on insect defences only is visible when low doses of conidia are applied, as too many conidia overwhelm the insect defences too quickly.

Additional comments

Minor comments:

It is more common to provide un-logged values of the LT50s and show these data in a table (Z x 10^Y).

---

## Round 0.2 · Minor Revisions

Dear authors, thanks for taking into consideration most comments of the reviewer. There are still a few comments from reviewer 1 that I would like you to address. I agree with reviewer 1 that Figure 1 should be moved to supplementary material. Please have a look to the data analysis suggestion and send back a new version of your manuscript.

Reviewer 1 ·

Basic reporting

The authors have addressed most of the comments or have coherently refuted the others.

Experimental design

Although the authors encourage to read Soth et al. (2022) for material and methods, I suggest that next time specify that cadavers are surface disinfected for mycosis observation. Otherwise the authors cannot be sure that the fungus comes from inside the insect.

Validity of the findings

No comment

Additional comments

No comment

Reviewer 2 ·

Basic reporting

The reviewed version of the manuscript titled “You are what you eat: fungal metabolites and host plant affect the susceptibility of diamondback moth to entomopathogenic fungi” (#74114) has been strongly improved following all the comments suggested by the reviewers, which has strengthened property its quality for publication. I strongly suggest accepting the manuscript for its publication. I only have some minor comments for its final improvement.

Experimental design

Regarding Kaplan-Meier analysis, what you are calculating is the mean survival time (MST). TL50 is more appropriate to be calculated with Probit. Please, check the concepts of Kaplan-Meier and adapt them appropriately.

Validity of the findings

The revision has covered all the questions suggested previously

Additional comments

Figure 1 is very clear and resolve many questions form the first revision. Anyway, I suggest including it as a supplementary figure.

---

## Round 0.3 · accepted · Accept

I think that the manuscript now is more clear and has improved.
Congratulations!